# The Applications of Artificial Intelligence in Cardiovascular Magnetic Resonance—A Comprehensive Review

**DOI:** 10.3390/jcm11102866

**Published:** 2022-05-19

**Authors:** Adriana Argentiero, Giuseppe Muscogiuri, Mark G. Rabbat, Chiara Martini, Nicolò Soldato, Paolo Basile, Andrea Baggiano, Saima Mushtaq, Laura Fusini, Maria Elisabetta Mancini, Nicola Gaibazzi, Vincenzo Ezio Santobuono, Sandro Sironi, Gianluca Pontone, Andrea Igoren Guaricci

**Affiliations:** 1University Cardiology Unit, Cardio-Thoracic Department, Policlinic University Hospital, 70121 Bari, Italy; adrianaargentiero92@gmail.com (A.A.); nicolo.soldato@gmail.com (N.S.); pabas2304@gmail.com (P.B.); eziosantobuono@gmail.com (V.E.S.); 2School of Medicine and Surgery, University of Milano-Bicocca, 20126 Milan, Italy; g.muscogiuri@gmail.com (G.M.); sandro.sironi@unimib.it (S.S.); 3Department of Radiology, IRCCS Istituto Auxologico Italiano, San Luca Hospital, 20149 Milan, Italy; 4Division of Cardiology, Loyola University of Chicago, Chicago, IL 60660, USA; mrabbat@lumc.edu; 5Radiologic Sciences, Department of Medicine and Surgery, University of Parma, 43126 Parma, Italy; chiaramartini10@gmail.com; 6Perioperative and Cardiovascular Imaging Department, Centro Cardiologico Monzino IRCCS, 20138 Milan, Italy; andrea.baggiano@cardiologicomonzino.it (A.B.); saima.mushtaq@ccfm.it (S.M.); laura.fusini@cardiologicomonzino.it (L.F.); maria.mancini@cardiologicomonzino.it (M.E.M.); gianluca.pontone@cardiologicomonzino.it (G.P.); 7Department of Cardiology, Azienda Ospedaliero-Universitaria, 43126 Parma, Italy; ngaibazzi@gmail.com; 8Department of Radiology, ASST Papa Giovanni XXIII Hospital, 24127 Bergamo, Italy; 9Department of Emergency and Organ Transplantation, University of Bari, 70121 Bari, Italy

**Keywords:** cardiac magnetic resonance, artificial intelligence, machine learning, deep learning

## Abstract

Cardiovascular disease remains an integral field on which new research in both the biomedical and technological fields is based, as it remains the leading cause of mortality and morbidity worldwide. However, despite the progress of cardiac imaging techniques, the heart remains a challenging organ to study. Artificial intelligence (AI) has emerged as one of the major innovations in the field of diagnostic imaging, with a dramatic impact on cardiovascular magnetic resonance imaging (CMR). AI will be increasingly present in the medical world, with strong potential for greater diagnostic efficiency and accuracy. Regarding the use of AI in image acquisition and reconstruction, the main role was to reduce the time of image acquisition and analysis, one of the biggest challenges concerning magnetic resonance; moreover, it has been seen to play a role in the automatic correction of artifacts. The use of these techniques in image segmentation has allowed automatic and accurate quantification of the volumes and masses of the left and right ventricles, with occasional need for manual correction. Furthermore, AI can be a useful tool to directly help the clinician in the diagnosis and derivation of prognostic information of cardiovascular diseases. This review addresses the applications and future prospects of AI in CMR imaging, from image acquisition and reconstruction to image segmentation, tissue characterization, diagnostic evaluation, and prognostication.

## 1. Introduction

Cardiovascular disease remains an integral field on which new research in both the biomedical and technological fields is based, as it remains the leading cause of mortality and morbidity worldwide [1]. Despite the progress of cardiac imaging techniques, the heart remains a challenging organ to study, secondary to respiratory motion, continuous cycles of contraction and relaxation, complex geometry, and variability of imaging protocols. Therefore, advanced tools are needed to support the use of cardiac imaging and aid physicians in daily cardiovascular practice [2,3,4,5,6,7,8]. Cardiovascular magnetic resonance (CMR) imaging allows very accurate evaluation of the function and structure of the heart chambers, with a high capability to handle complex diagnostic issues [9]. Artificial intelligence (AI) has emerged as one of the major innovations in the field of diagnostic imaging [10,11,12]. Recent advances in machine learning (ML) techniques for the management of workflow, optimization of image acquisition methods, and evaluation of images have opened up new avenues in cardiovascular practice. Automated quantitative assessments aimed at the detection, characterization, and monitoring of diseases are now possible. In the future, AI will be increasingly present in daily clinical practice, aiding the clinician in the diagnosis of, and derivation of prognostic information on, cardiovascular diseases [13,14].

## 2. Artificial Intelligence

Artificial intelligence is the aim to develop computers with human intelligence [15,16]. Machine learning (ML) and deep learning (DL) are two subcategories of artificial intelligence. ML uses an algorithm in which the system adapts only after receiving human feedback. A prerequisite for the use of technology is the existence of structured data. The system is first fed with structured and categorized data and then understands how to classify the new data by type. Based on the classification, the system then performs scheduled tasks. After an initial phase of the application, the algorithm is optimized by human feedback, which indicates to the system the wrong and correct categorizations. In the case of DL, structured data are not necessary [17,18,19]. DL systems work in multilayer neural networks, which combine several algorithms and are modeled on the human brain. This allows the system to process even unstructured data. 

## 3. Machine Learning

ML algorithms can be subdivided into supervised, unsupervised, and reinforced learning [20,21]. In supervised learning, the machine is equipped with a labeled dataset. It already has input and output parameters. Then, when a new dataset is supplied to the machine, the supervised learning algorithm examines the data and produces the correct output based on the labeled data. In unsupervised learning, the machine has no labeled dataset; the algorithm is designed to try to learn on its own without any supervision of the data. This involves the grouping of data. In reinforcement learning, the algorithms are designed in such a way that the machine tries to find an optimal solution, adopts the principle of reward and punishment, and with this approach moves towards the correct result. The most commonly used ML techniques in the field of cardiac imaging and diagnosis are logistic regression, support vector machines, random forests, cluster analysis, artificial neural networks, and convolutional neural networks [22,23,24,25] (Figure 1). 

### 3.1. Logistic Regression

Logistic regression is a supervised machine learning technique. It is based on the use of the logistic function (sigmoid), which converts real values to a value between 0 and 1. In the training phase, the algorithm receives a training dataset consisting of N examples. Each example consists of attributes X and a label Y indicating the correct classification. At the end of the training, the algorithm produces a model that can be used to classify any other example not included in the training set. 

### 3.2. Support Vector Machine 

Support vector machines (SVMs) are supervised ML models segregating the data into two or more classes, obtaining a linear, binary, nonprobabilistic classifier. An SVM model is a representation of the examples as points in space, mapped in such a way that the examples belonging to the two different categories are separated by as large a space as possible. New examples are then mapped in the same space, and the prediction of the category to which they belong is made based on the side in which they fall. In addition to linear classification, it is possible to make use of SVMs to effectively carry out nonlinear classification using the kernel method by implicitly mapping their inputs into a multidimensional characteristics space.

### 3.3. Random Forest

The random forest (RF) method is an ensemble learning method for classification, a regression that operates by building a multitude of decision trees during training. For classification activities, the random forest output is the class selected by most trees. For regression activities, the mean or mean forecast of the individual trees is returned. 

### 3.4. Cluster Analysis 

Cluster analysis finds subgroups within the input data in an unsupervised manner. Clustering algorithms group elements based on their mutual distance, and therefore, whether or not they belong to a set depends on how far the element under consideration is from the set itself.

### 3.5. Artificial Neural Network 

Artificial neural network (ANN) is a computational model composed of artificial “neurons”, vaguely inspired by the simplification of a biological neural network, determined by the propagation of input data through a nonlinear transformation network. 

### 3.6. Convolutional Neural Network

Convolutional neural networks (CNN) are the backbone of DL applications. A convolutional neural network is a type of feed-forward artificial neural network in which the connectivity pattern between neurons is inspired by the organization of the animal visual cortex, the individual neurons in which are arranged in such a way as to respond to the overlapping regions that tessellate the visual field.

## 4. Deep Learning

Deep learning (DL) evolved from ML (Figure 2). It works in a layered architecture and uses the artificial neural network, a concept inspired by the biological neural network. It simply takes the data connections among all the artificial neurons and adjusts them based on the data model. More neurons are needed if the data size is large. It automatically presents learning at multiple levels of abstraction, thus allowing a system to learn the mapping of complex functions without depending on any specific algorithm. In DL, the accuracy of the output depends on the amount of data. It consists of three layers:Input layer: the input layer is used to take input data from sources and then pass it to the hidden layers of the neural network. It does not perform any calculations.Hidden level: this level consists of many hidden levels. All the calculation is performed at this level. After all the calculations are complete, it proceeds to the output level.Output level: this level is used to provide the output to the outside world.

## 5. Current Applications of Artificial Intelligence

Machine learning will have an impact on all aspects of CMR, from patient programming to image analysis and prognosis. Optimal CMR imaging requires correct patient positioning and precise image planning [26]. The acquisition and reconstruction of images is a task typically entrusted to human experience that can now be automated with AI and reduce time. Machine learning methods have been used to optimize frequency regulation for 3-Tesla CMR and for automatic correction of artifacts. Recently, DL techniques have been shown to offer superior performance in terms of reconstruction quality as well as offering high efficiency [27]. Although CMR imaging offers many benefits for assessing cardiac structure and function, CMR image acquisition is time consuming because of complex cardiac and respiratory movements. Thus, reducing scan time has always been one of CMR imaging’s biggest challenges. Over the past decade, methods such as parallel imaging and compressed sensing (CS) have been increasingly used to accelerate CMR image acquisition (Figure 3 and Figure 4) [28,29,30]. 

## 6. Image Acquisition

Several AI solutions have been proposed by different vendors in terms of image acquisition and reconstruction (Table 1) [31,32]. The main goals of all these approaches are to simplify the image acquisition, facilitate CMR acquisition, and in some cases improve image quality, reduce the time of acquisition, and improve overall efficiency [27]. Forman et al. studied a scan protocol for coronary magnetic resonance angiography (CMRA) based on multiple breath holds featuring 1D motion compensation and compared the resulting image quality with that from a navigator-gated free-breathing acquisition. In other work, the investigators used iterative reconstruction with self-navigated free-breathing CMRA for retrospective reduction of respiratory motion artifacts (Figure 5) [33,34]. Nakamura et al. demonstrated that noncontrast CMRA using CS could greatly shorten acquisition time compared with that of conventional navigator-gated coronary MRA while maintaining acceptable visualization at 3T [35].

### 6.1. Slice Position

Automatic slice positioning was one of the first applications of AI during image acquisition [36]. Frick et al. described a fully automated approach in which slice position was detected using a deformation and segmentation algorithm. Subsequently, several algorithms have been developed that were able to identify cardiac landmarks and automatically plan the cardiac planes [37,38,39]. Lu et al. proposed an algorithm that was able to identify landmarks and prescribe long and short axis views from a 3D acquisition [40]. The authors created a model that was subsequently adapted to the heart of the patient and allowed prescribing the slices [40]. A more recent approach was proposed by Blansit et al., who developed an algorithm based on DL training on hundreds of annotated landmarks on CMR [41]. These approaches, based on training of 2D datasets, represent the basis for several vendors and may allow the ability to prescribe planes with low variability compared with those prescribed via manual acquisition. 

### 6.2. Image Quality

Improvement of image quality has been developed by several vendors by decreasing the noise of pictures (Figure 6). The “Deep Resolve” algorithm is a deep learning reconstruction able to decrease the noise of images. It is divided into “sharp” and “gain” reconstructions. In “Deep Resolve Gain” acquisition, the algorithm, starting from raw data, identifies the parts of images with more noise and increases the noise reduction in these parts [47]. In “Deep Resolve Sharp” acquisition the images enter the image reconstruction in a “high-resolution mode” while they are acquired with low resolution. The algorithm was trained on high-resolution images, so in the presence of low resolution pictures, the algorithm decreases the noise, simulating an image with high resolution from the raw data in order ensure the quality. 

AIR Recon DL reconstruction is another technique that is able to increase signal-to-noise ratio, reduce truncation artifacts, and increase spatial resolution compared with standard reconstruction [42]. Using the AIR DL reconstruction, images can be reconstructed using “low”, “medium”, or “high” filters. The algorithm was created using millions of images, gradient backpropagation, and the ADAM optimizer. Compared with the previous algorithm, AIR DL has been evaluated in CMR [32]. Van der Velde et al. analyzed the impact of AIR DL on late gadolinium enhancement (LGE) images, demonstrating that it was possible to obtain LGE images with decreased noise [43]. Similar results were observed by Muscogiuri et al., who showed that it was possible to obtain images from multisegment LGE with the same image quality as those from standard single-segment LGE [32].

### 6.3. Image Speed Acquisition 

Cine images in CMR result from a compromise between temporal and spatial resolution; in particular, decreased time of acquisition may cause image quality impairment [48]. However, compressed sensing provides the possibility of decreasing the time of cine image acquisition [49]. Hauptman et al. trained a U-Net algorithm that was able to remove aliasing and provide similar volumes in cine images thirteen times faster than standard cine images [44]. Sandino et al. developed another model, called “DL-ESPIRIT”, that was able to combine the CNN reconstruction with standard reconstruction and provide similar results to standard CS images, accelerating the time of acquisition by twelve times [45]. Kustner et al. developed another approach, a 4D DL-based reconstruction algorithm for 3D Cartesian cine data that was called CINENet [46]. In particular, the CINENet algorithm was able to acquire the left ventricular volume in 15 s, providing the same result as cine images [46]. 

## 7. Image Segmentation

The delineation of the contours of the heart chambers and myocardium (segmentation) is the first step in the processing of CMR images (Table 2 and Figure 7) [50], as the quantitative parameters of left ventricular end-diastolic volume (LVEDV), left ventricular end-systolic volume (LVESV), right ventricular end diastolic volume (RVEDV), right ventricular end-systolic volume (RVESV), and ejection fraction (EF) are derived from it and it has a role in prognostication [51,52].

Romaguera et al. used a CNN for segmentation of short-axis CMR images, while Bernard et al. showed how using ML methods in segmentation could give accurate results [53,54]. Automated segmentation methods based on ML and DL have been developed, but occasionally, manual correction is still necessary. As seen in the works on the use of artificial intelligence in the segmentation process, is more difficult to delineate the contours of the right ventricle than those of the left ventricle because of the right ventricle’s smaller wall thickness and irregular shape, with the presence of greater trabeculae [62,63,64].

Bai et al. showed that an automatic method based on the use of DL in the process of segmentation and measurement of quantitative parameters in CMR imaging presented a performance equal to that of human experience [55]. Similar results were shown by Penso et al., who demonstrated a good correlation between volume calculated with DL and that calculated with a manual approach [56]. Atrial segmentation could be useful for management of atrial fibrillation, in particular for the planning of atrial fibrillation ablation both in the preoperative period and in follow-up. Xiong et al. developed a CNN on 3D LGE MRI to automatically segment the left atrium [57]. Using DL and manual segmentation, Yang et al. demonstrated the ability of atrial scar segmentation [58]. In parallel, it is well known that the right atrial (RA) area predicts mortality in patients with pulmonary hypertension and is recommended by the European Society of Cardiology/European Respiratory Society pulmonary hypertension guidelines. Importantly, the advent of deep learning may allow more reliable measurement of RA areas in order to improve clinical assessments [65]. Xu et al. proposed a CNN-based automatic segmentation method in noncontrast cine MR images of myocardial infarction areas; this method obtained high consistency with human experience and LGE images [61]. 

## 8. Myocardial Tissue Characterization

ML methods have been applied for the automatic quantification of LGE, overcoming the limits related to its irregular and multifocal appearance, the variation of gadolinium kinetics, and the variability among different centers in regard to accuracy and reproducibility (Table 3 and Figure 8) [59,60,66,67]. ML has also been applied to cardiac relaxometry, specifically T1 mapping, which is useful for identifying changes in extracellular volume [68,69,70]. Radiomics, an ML technique capable of handling high-dimensional data [71], refers to analysis of medical images aimed at obtaining quantitative information, through appropriate mathematical methods and the use of computers, that cannot be detected by simple visual observation by an operator. Various features can be extracted from images, the most important being intensity- and texture-based morphological features (summarized in the term “texture analysis” (TA)) [72]. TA refers to any geometric and/or repetitive arrangement of grey levels and allows the segmentation, analysis, and classification of medical images according to the underlying tissue structure, thus offering the possibility to overcome the limitations of pure interpretation of visual images [73,74]. There are several applications of radiomics and TA: to detect myocardial fibrosis in patients with hypertrophic cardiomyopathy (HCM), to differentiate healthy controls from patients with cardiomyopathy (HCM, amyloid), and to perform scar segmentation in myocardial infarction (MI) in the differential diagnosis between acute and chronic infarction [75,76,77,78,79]. In regard to the application of TA in T1–T2 mapping, Neisius et al. applied it to discriminate between HCM and hypertensive heart disease, while Baessler et al. demonstrated its diagnostic accuracy in both acute infarct-like myocarditis and chronic myocardial inflammation/myocarditis [80,81].

## 9. Diagnosis

ML can help physicians in the accurate and early image-based diagnosis of cardiovascular disease. Several papers have used conventional imaging indices as input data to diagnose various heart diseases such as HCM, DCM, MI, and ARVC (Table 4).

### 9.1. Myocardial Infarction

Machine learning methods offer simplification and acceleration of the diagnostic path of MI and are useful as a guide for treatment strategies [88]. Baessler et al. used CMR images with contrast as a reference to differentiate chronic from subacute MI on noncontrast CMR images, while Zhang et al. directly used noncontrast CMR images to diagnose chronic MI [74,85].

### 9.2. Cardiomyopathies

Machine learning is an excellent method that allows for the differentiation of various cardiomyopathies [88]. Gopalakrishnan et al. used CMR parameters of the left ventricle, right ventricle, and overall heart from a cohort of 83 pediatric subjects in order to characterize five different cardiomyopathies: HCM, DCM, ARVC, left ventricle noncompaction, and myocarditis [86]. Khened et al. and Wolterink et al. used conventional CMR indices as input for the classification of subjects into five categories (healthy, HCM, DCM, ARVC, and MI) and obtained accuracies of 0.96 and 0.86, respectively, on the test set [62,63,83,84]. Focusing on radiomic features, Nelsius et al. demonstrated an ML model for differentiating HCM from hypertensive heart disease using radiomic features derived from T1 mapping sequences [80].

### 9.3. Heart Failure

Heart failure (HF) is a clinical syndrome due to a structural and/or functional abnormality of the heart that results in elevated intracardiac pressures and/or inadequate cardiac output at rest and/or during exercise. Because of the progressive aging of the general population, heart failure has assumed an increasingly relevant epidemiological problem and currently represents the cardiovascular disease with the greatest increase in incidence. Baessler et al. applied texture analysis on cardiac MRI T1 and T2 mapping and obtained quantitative imaging parameters for the diagnosis of acute or chronic heart failure [82]. Moreno et al. used a dataset that contained cine-MRI images (HF with infarction having an EF < 40%, HF without infarction having an EF < 40%, hypertrophy having a normal EF > 55%, and normal patients having EF > 55%) to describe and characterize heart motion patterns along the cardiac cycle through SVM and RF [89].

### 9.4. Abnormal Wall Motion

Machine learning facilitates postprocessing and image analysis for the characterization of wall motion [88]. Mantilla et al. and Afshin et al. proposed an ML method for classifying normal/abnormal wall motion in left ventricular function using CMR [87,90].

## 10. Prognosis

Cardiac magnetic resonance has demonstrated the ability to prognosticate various cardiovascular conditions [6,91,92,93,94,95,96,97,98]. Cheng et al. showed that greater heterogenicity of LGE in patients with HCM and systolic dysfunction was associated with adverse events [99]. Through the use of biomarkers derived from imaging, ML methods may add value beyond traditional risk scores in the prediction of adverse cardiovascular events, as proposed by the Framingham Heart Study and other cohorts. In the MESA (Multiethnic Atherosclerosis Study) study, there was better prediction of risk, with greater accuracy, in the prediction of cardiac events. [100]. In addition, ML may predict cardiac arrhythmias in patients who have survived heart attacks [101] and therefore have a discriminating power in risk stratification, similar to that of EF and scar size, using the characteristics, position, and size of the scar derived from CMR. It may also be useful in predicting a favorable response to cardiac resynchronization therapy using ECG, clinical, and heart motion analysis data [102].

## 11. Limitations

Although machine learning methods have proven effective and have been validated, there are practical difficulties in implementing them in the clinical setting for several reasons, including data quality. Homogenization of clinical data and imaging protocols from different centers is an essential element to check before using datasets as input into a standard ML model. In addition, since clinical data add critical information to images, it is essential to integrate clinical data in electronic medical records to the imaging dataset. Cost is another critical issue, as substantial investment would be required to develop more complex ML methods. Although DL has shown promising results for image processing in the context of high-dimensional datasets and supported by the possibility of real-time reconstruction, in the use of powerful computational algorithms, additional limitations must be recognized. The lack of large, publicly available CMR datasets that can be used to objectively compare different algorithms; the lack of generalization capabilities when previously unseen samples are presented; the difficulties with rare diseases (and, conversely, rare presentations of common entities such as congenital heart disease); and the “black-box” nature of DL algorithms, in that it is often unclear what information is used to arrive at a particular classification or result, represent just a few of the real limitations of this intriguing technique.

## 12. Future Perspectives

In this review, we report the use of ML in all aspects of CMR, from the acquisition and reconstruction of images to the derivation of prognostic information. Despite the significant advances described above, there is a need for controlled prospective clinical trials in which ML techniques are compared with human evaluation of magnetic resonance datasets to establish the effectiveness of such methods in clinical practice. The algorithms must be validated using other cohorts besides that under consideration, including data coming from different centers and from different acquisition devices. Furthermore, to compare the performance of different algorithms, it would be important to evaluate the efficiency of each algorithm on a common dataset. Finally, to enable progress in this field, publicly accessible software would be required to allow other groups to run, study, modify, and thus improve software and monitor potential bias in the algorithm [103]. An inevitable and widespread concern is data privacy, so it would be necessary to build a privacy protection algorithm that combines encryption and AI methods in order to obtain fast, secure, and generalizable models. As larger datasets become available, predictive models for both disease progression and simulated response to therapy are expected to develop.

## 13. Conclusions

The advancement of techniques in the field of AI can be attributed both to the fact that AI is able to manage high-dimensional data, integrating information from multiple fields, and to the increasing availability of data through mobile applications and data transformation from the healthcare system into digital form. However, the applicability of these approaches in cardiovascular applications remains limited because of the intrinsic peculiarities of cardiac imaging, studies based on small samples that involve a risk of overfitting, and the lack of interoperability of many systems used. However, the use of AI in cardiovascular medicine could pave the way towards better diagnosis and precision medicine and revolutionize the monitoring and treatment of individual diseases.

## Figures and Tables

**Figure 1 jcm-11-02866-f001:**
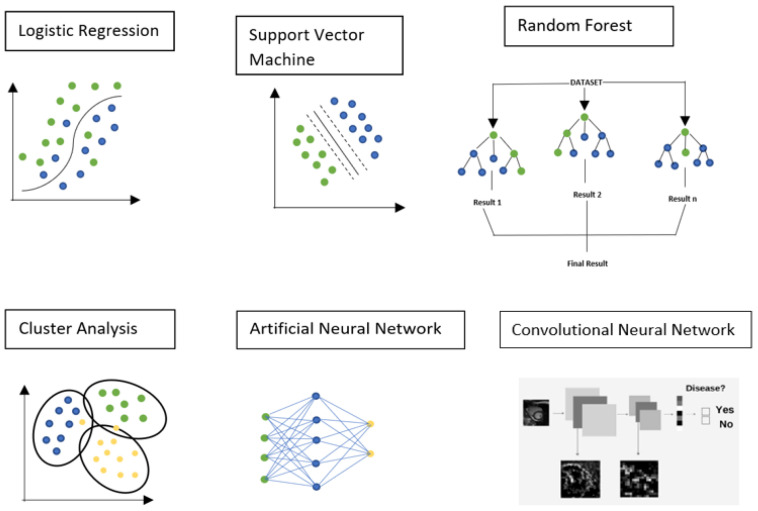
The most commonly used Machine Learning (ML) techniques in the field of cardiac imaging.

**Figure 2 jcm-11-02866-f002:**
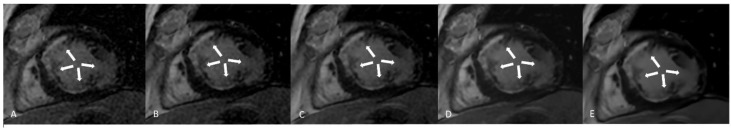
LGE sequences acquired using an artificial intelligence reconstruction deep learning algorithm. Forty-one-year old patient with previous myocardial infarction on anterior, anteroseptal, inferoseptal, inferior, and inferolateral segments (arrows, (**A**–**E**), respectively). Image noise decreased progressively with increase in AIRDL reconstruction in both 2D-SSLGE ((**A**): 2D-SSLGE AIRDL 0%, (**B**): 2D-SSLGE AIRDL 25%, (**C**): 2D-SSLGE AIRDL 50%, (**D**): 2D-SSLGE AIRDL 75%, (**E**): 2D-SSLGE AIRDL 100%). 2D-SSLGE—2D single segmented inversion recovery gradient echo late gadolinium enhancement sequences; AIRDL—artificial intelligence reconstruction deep learning.

**Figure 3 jcm-11-02866-f003:**
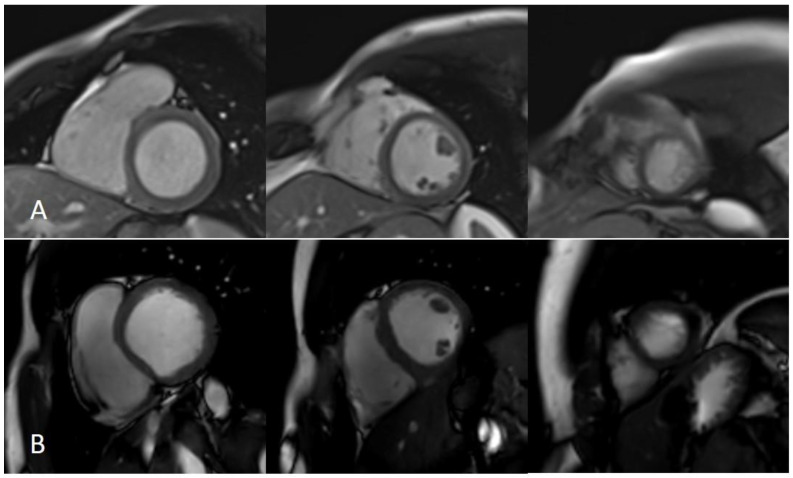
Short axis acquired using parallel imaging and compressed sensing methods. (**A**) string shows a functional cardiac plane acquired using parallel imaging with a 1.5T MR system, while (**B**) string shows an SA acquired using the CS method with a 3T MR system. SA—short axis; CS—compressed sensing.

**Figure 4 jcm-11-02866-f004:**
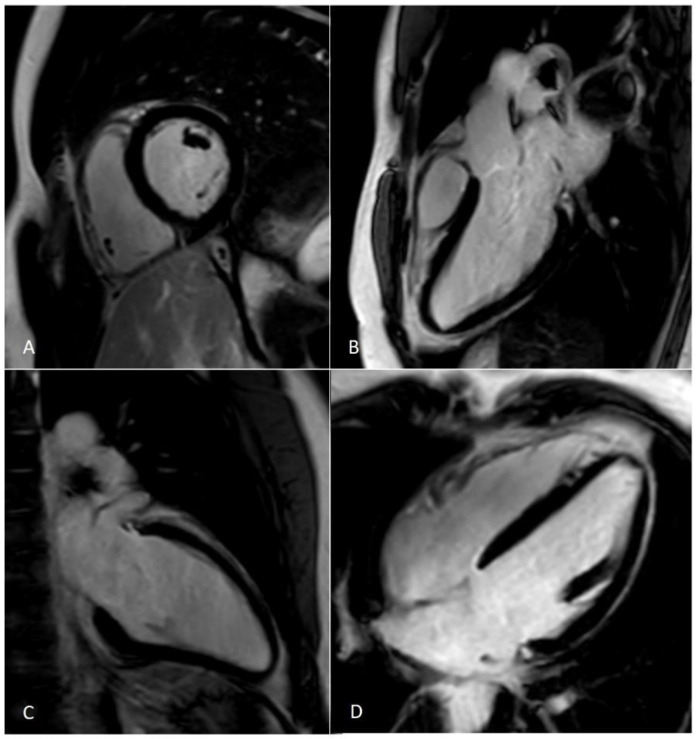
LGE —late gadolinium enhancement sequences acquired using the compressed sensing technique. Panel (**A**)—Left ventricle (LV) and right ventricle (RV) short axis view at the level pf the papillary muscles; (**B**)—LV three chamber view; (**C**)—LV two chamber view; (**D**)—LV and RV four chamber view.

**Figure 5 jcm-11-02866-f005:**
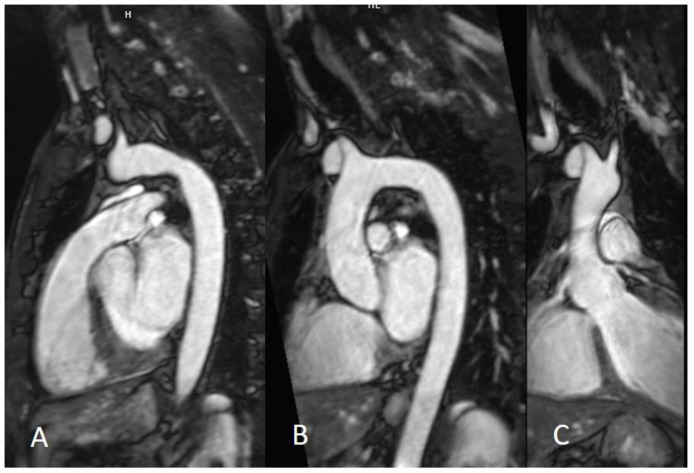
Three-dimensional navigator whole-heart CMRA sequence. The 3D CMRA allows to acquire the whole heart in a gated free-breathing acquisition (**A**), with the possibility of subsequent MPR reconstructions (**B**,**C**). CMRA—coronary magnetic resonance angiography; MPR—Multiplanar reconstruction.

**Figure 6 jcm-11-02866-f006:**
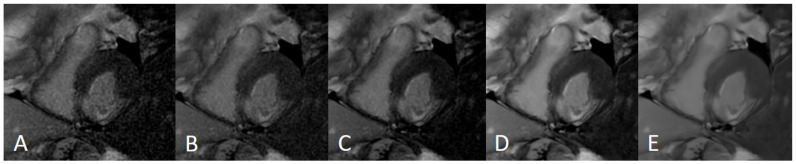
Artifacts reduction with artificial intelligence implementation. Eighty-two-year-old male patient with previous inferior and inferolateral myocardial infarction. Image (**A**–**C**) show the reconstruction of 2D-MSLGE with NR 0% (**A**), NR 25% (**B**), and NR 50% (**C**), respectively. The increasing percentage in NR reconstruction yielded a progressive reduction in image noise in 2D-MSLGE starting from NR 0% (**C**) and moving through NR 25% (**D**) and NR 50% (**E**). A breath artifact characterizing the inferior and inferolateral midapical segments was reduced in the reconstruction in which the 100% artificial intelligence algorithm was applied. In fact, a reduction in quantum noise resulted in better contrast resolution. 2D-MSLGE—2D multisegment late gadolinium enhancement; NR—artificial intelligence reconstruction deep learning noise reduction.

**Figure 7 jcm-11-02866-f007:**
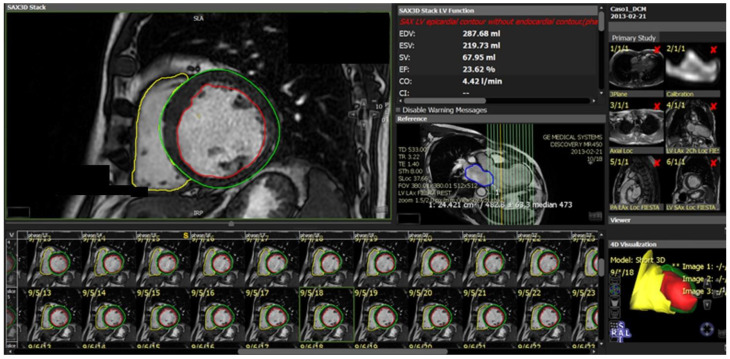
Image segmentation platform. Functional and volume analysis to obtain ejection fraction, volumes, stroke volume, cardiac index, and left ventricle mass in end diastolic phase. Green: epicardial contour; red: endocardial contour; yellow: right ventricle. EDV—end diastolic volume; ESV—end systolic volume; SV—stroke volume; EF—ejection fraction; CO—cardiac output; CI—cardiac index.

**Figure 8 jcm-11-02866-f008:**
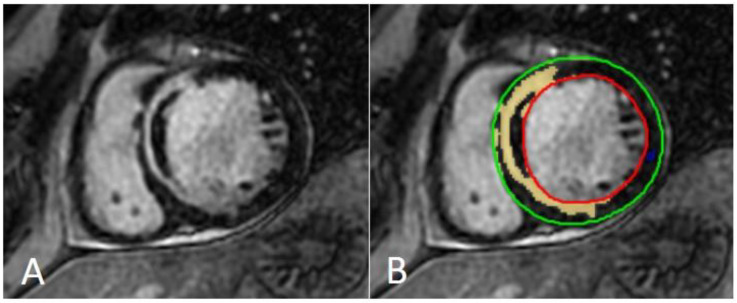
Assessment of myocardial scar with semiautomatic tissue characterization. Semiautomatic tissue characterization algorithm allowing the identification of myocardial scars (represented by hyper-enhanced myocardium in panel (**A**) and yellow-colored myocardium in panel (**B**)) by positioning a region of interest within the territory of the remote/normal myocardium (dark myocardium). Green: epicardial contour; red: endocardial contour; yellow: myocardial scar; blue: normal myocardium.

**Table 1 jcm-11-02866-t001:** Machine learning and deep learning for image acquisition and reconstruction.

	Method	Image Substrate	Application
Muscogiuri et al. (2021) [32]	DL	2D multisegment late gadolinium enhancement	Noise reduction
Forman et al. (2015) [33]	CS	Free-breathing whole-heart coronary MRA	Reduction of respiratory motion artifacts
Forman et al. (2014) [34]	CS	High-resolution 3D whole-heart coronary MRA	Shortening of acquisition time
Schemper et al. (2018) [27]	CNN	Cine	Automatic reconstruction
Frick et al. (2011) [36]	ML	CMR imaging	Automatic view planning
Yokoyama et al. (2015) [37]	ML	CMR imaging	Automatic slice alignment method
Nitta et al. (2013) [38]	ML	CMR imaging	Automatic slice alignment method
Oktay et al. (2017) [39]	ML	Cine	Localization of anatomical landmarks
Lu et al. (2011) [40]	ML	CMR imaging	Automatic view planning
Blansit et al. (2019) [41]	DL	CMR imaging	Localizaion of anatomical landmarks
Lebet et al. (2020) [42]	CNN	CMR imaging	Improvement of image quality
Van Der Velde et al. (2021) [43]	DL	LGE	Improvement of image quality
Hauptmann et al. (2019) [44]	CNN	CMR imaging	Shortening of reconstruction time and improvement of image quality
Sandino et al. (2021) [45]	DL	Cine	Shortening of reconstruction time and improvement of image quality
Kustner et al. (2020) [46]	DL	Cine	Shortening of reconstruction time and improvement of image quality

ML—machine learning; DL—deep learning; CS—compressed sensing; CNN—convolutional neural network; CMR—cardiac magnetic resonance imaging; CMRA—coronary magnetic resonance angiography; LGE—late gadolinium enhancement.

**Table 2 jcm-11-02866-t002:** Machine learning and deep learning for image segmentation.

	Method	Image Substrate	Application
Romaguera et al. (2018) [53]	CNN	CMR imaging	Ventricular segmentation
Bernard et al. (2018) [54]	DL	CMR imaging	Ventricular segmentation
Bai et al. (2018) [55]	DL	CMR imaging	Ventricular segmentation
Penso et al. (2021) [56]	DL	CMR imaging	Ventricular segmentation
Xiong et al. (2019) [57]	CNN	LGE	Atrial segmentation
Yang et al. (2018) [58]	DL	LGE	Atrial scar segmentation
Zabihollahy et al. (2019) [59]	DL	LGE	Myocardial scar segmentation
Moccia et al. (2019) [60]	DL	LGE	Myocardial scar segmentation
Xu et al. (2018) [61]	CNN	Cine	Myocardial infarction area segmentation

DL—deep learning; CNN—convolutional neural network; CMR—cardiac magnetic resonance; MRI— magnetic resonance imaging; LGE—late gadolinium enhancement.

**Table 3 jcm-11-02866-t003:** Machine learning and deep learning for myocardial tissue characterization.

Author	Method	Image Substrate	Application
Fahmy et al. (2018) [66]	CNN	LGE	Segmentation and quantification of scar volume in patients with HCM
Hann et al. (2018) [70]	DL	T1 mapping	Automated LV segmentation of T1 maps in order to speed up LGE quantification based on T1 mapping
Thornhill et al. (2014) [79]	Radiomics and TA	LGE	Detection of myocardial fibrosis in patients with HCM
Schofield et al. (2019) [75]	Radiomics and TA	Cine	Differentiation among several causes of myocardial hypertrophy (HCM, amyloid, and aortic stenosis) and healthy controls
Engan et al. (2010) [76]	Radiomics and TA	LGE	Discrimination of patients with low and high risk of arrhythmias
Kotu et al. (2013) [77]	Radiomics and TA	LGE	Automated segmentation of scarred tissue areas
Larroza et al. (2017) [78]	Radiomics and TA	LGE, Cine	Differential diagnosis between acute and chronic infarction
Neisius et al. (2019) [80]	Radiomics and TA	Native T1 mapping	Discrimination between hypertrophic cardiomyopathy and hypertensive heart disease
Baessler, et al. (Radiology 2018 Nov) [81]	Radiomics and TA	Native T1–T2 mapping	Diagnostic accuracy in acute infarct-like myocarditis
Baessler, et al. (Radiology 2018 Jan) [82]	Radiomics and TA	Native T1–T2 mapping	Diagnostic accuracy in chronic myocardial inflammation/myocarditis

DL—deep learning; CNN—convolutional neural network; TA—texture analysis; HCM—hypertrophic cardiomyopathy; LGE—late gadolinium enhancement.

**Table 4 jcm-11-02866-t004:** Machine learning and deep learning for diagnosis.

Author	Method	Image Substrate	Myocardial Disease
Khened et al. (2018) [83]	CNN	Cine	HCM, DCM, MI, and ARVC
Ammar et al. (2021) [84]	CNN	Cine	HCM, DCM, MI, and ARVC
Neisius et al. (2019) [80]	Radiomics and TA	Native T1 maps	Discrimination between HCM and hypertensive heart disease
Baessler et al. (Radiology 2018 Jan) [74]	Radiomics and TA	Cine	Differentiation of chronic from subacute MI
Zhang et al. (2019) [85]	DL	Cine	Chronic MI
Gopalakrishnan et al. (2015) [86]	ML		HCM, DCM, ARVC, LVNC, and myocarditis
Wolterink et al. (2018) [62]	RF	Cine	Healthy, HCM, DCM, ARVC, and MI
Snaauw et al. (2019) [63]	CNN		Healthy, HCM, DCM, ARVC, and MI
Baessler et al. (Radiology 2019) [82]	Radiomics and TA	Native T1–T2 mapping	Acute or chronic heart failure-like myocarditis
Mantilla et al. (2013) [87]	ML	Cine	Abnormal wall motion

DL—deep learning; CNN—convolutional neural network; TA—texture analysis; RF—random forest; ML—machine learning; HCM—hypertrophic cardiomyopathy; DCM—dilated cardiomyopathy; ARVC—arrhythmogenic right ventricular cardiomyopathy; LVNC—left ventricle noncompaction; MI—myocardial infarction.

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
