# Peer review of "The Applications of Artificial Intelligence in Cardiovascular Magnetic Resonance—A Comprehensive Review"

_jcm, 2022, doi:10.3390/jcm11102866_

Round 1

Reviewer 1 Report

The review by Adriana Argentiero and colleagues entitled “THE APPLICATIONS OF ARTIFICIAL INTELLIGENCE IN CARDIOVASCULAR MAGNETIC RESONANCE –A Comprehensive Review” aimed to address the applications and future prospects of AI in CMR imaging from image acquisition and reconstruction, to image segmentation, tissue characterization, diagnostic evaluation and prognostication.

The abstract summarizes the general significance of the manuscript and the review leads some evidence to such point; however, some major issues need to be addressed to improve the significance of the manuscript:

-“Artificial Intelligence” paragraph should be integrated with some images and some summary diagrams. Moreover, the role of AI in the cardiovascular field should be stressed, and this article should be cited: “Visco V, Ferruzzi GJ, Nicastro F, et al. Artificial Intelligence as a Business Partner in Cardiovascular Precision Medicine: An Emerging Approach for Disease Detection and Treatment Optimization. Curr Med Chem. 2021;28(32):6569-6590. doi:10.2174/0929867328666201218122633”.

-Furthermore, the revision should be made more homogeneous, avoiding a simple list of the various studies.

Finally, the bibliography is not complete. In particular, the right atrial (RA) area predicts mortality in patients with pulmonary hypertension and is recommended by the European Society of Cardiology/European Respiratory Society pulmonary hypertension guidelines. The advent of deep learning may allow more reliable measurement of RA areas to improve clinical assessments; consequently, this article should be cited: “Alandejani F, Alabed S, Garg P, et al. Training and clinical testing of artificial intelligence derived right atrial cardiovascular magnetic resonance measurements. J Cardiovasc Magn Reson. 2022;24(1):25. Published 2022 Apr 7. doi:10.1186/s12968-022-00855-3”.

Another essential point is the assessment of right ventricular size and function from CMR images using artificial intelligence; subsequently, this article should be cited: “Wang S, Chauhan D, Patel H, et al. Assessment of right ventricular size and function from cardiovascular magnetic resonance images using artificial intelligence. J Cardiovasc Magn Reson. 2022;24(1):27. Published 2022 Apr 11. doi:10.1186/s12968-022-00861-5”.

Minor issues:

-The year for each study should be included in Tables.

Author Response

Manuscript ID: jcm-1722265
Type of manuscript: Review
Title: THE APPLICATIONS OF ARTIFICIAL INTELLIGENCE IN CARDIOVASCULAR MAGNETIC RESONANCE – A Comprehensive Review

REVIEWER N° 1 

The authors express their gratitude for the comments and appropriate suggestions. Revision has been made accordingly. Referring to the specific points: 

Q1: “Artificial Intelligence” paragraph should be integrated with some images and some summary diagrams. Moreover, the role of AI in the cardiovascular field should be stressed, and this article should be cited: “Visco V, Ferruzzi GJ, Nicastro F, et al. Artificial Intelligence as a Business Partner in Cardiovascular Precision Medicine: An Emerging Approach for Disease Detection and Treatment Optimization. Curr Med Chem. 2021;28(32):6569-6590. doi:10.2174/0929867328666201218122633”.
A1: We agree with the reviewer and we have integrated the manuscript with a new figure, new comments and the new reference (new ref. 12)

Q2: Furthermore, the revision should be made more homogeneous, avoiding a simple list of the various studies. Finally, the bibliography is not complete. In particular, the right atrial (RA) area predicts mortality in patients with pulmonary hypertension and is recommended by the European Society of Cardiology/European Respiratory Society pulmonary hypertension guidelines. The advent of deep learning may allow more reliable measurement of RA areas to improve clinical assessments; consequently, this article should be cited: “Alandejani F, Alabed S, Garg P, et al. Training and clinical testing of artificial intelligence derived right atrial cardiovascular magnetic resonance measurements. J Cardiovasc Magn Reson. 2022;24(1):25. Published 2022 Apr 7. doi:10.1186/s12968-022-00855-3”.

A2: We agree with the reviewer and we tried to rewrite some parts and to update some papers cited in the manuscript’s tables. Moreover, we followed the suggestion of the Reviewer adding the new reference (new ref. 64).

Q3: Another essential point is the assessment of right ventricular size and function from CMR images using artificial intelligence; subsequently, this article should be cited: “Wang S, Chauhan D, Patel H, et al. Assessment of right ventricular size and function from cardiovascular magnetic resonance images using artificial intelligence. J Cardiovasc Magn Reson. 2022;24(1):27. Published 2022 Apr 11. doi:10.1186/s12968-022-00861-5”.

A3: We agree with the Reviewer and followed the suggestion (new ref. 57).

Q4: Minor issues: -The year for each study should be included in Tables.

A4: We have modified in agreement.

All the Best,

Andrea Igoren Guaricci, MD, PhD

Reviewer 2 Report

It is an interesting review on an up-to-date topic regarding the use of artificial intelligence in CMR both in terms of image acquisition and analysis. The broader use of AI in CMR is becoming a reality. I have some comments, which in my opinion would improve the manuscript:

Major issues:

  1. The division into various categories presented in the tables seems rather arbitrary. It would profit from a more ordered approach. It is often not followed by text description. Many citations in the text are not found in the tables. Also I do not see the difference between scar segmentation and delineation (as presented separately in Table 2 and 3). But it refers to other tables too. Please reconsider the construction of the tables and their connection to text.
  2. Line 236 - for both LV and RV - usually it is far easier to apply AI to LV segmentation than RV analysis? Please discuss this problem separately. 
  3. Line 157 - aortic stenosis is not a cardiomyopathy - please correct
  4. Line 300 - normal contractile pump function refers to systolic heart failure and not all heart failure category (not diastolic) - please adjust accordingly.
  5. Please add a paragraph on current limitations of AI use in CMR and on the applications which are currently commercially available. 

Minor issues:

Line 141 - please remove F - there is no such panel on Figure 1.

Figure 6 label - assessment of...

Figure 7 - there are only panels A-E presented, but in the text panel F and G are also described - please correct

Line 284 - uin?

Line 294 - please remove capital letters from the disease names. Also please change ARVD to ARVC or AC throughout the text as it is the correct abbreviation, also in Line 296 there is only ARV

Line 308 - please use bold

Author Response

Manuscript ID: jcm-1722265
Type of manuscript: Review
Title: THE APPLICATIONS OF ARTIFICIAL INTELLIGENCE IN CARDIOVASCULAR MAGNETIC RESONANCE – A Comprehensive Review

REVIEWER N° 2 

The authors express their gratitude for the comments and appropriate suggestions. Revision has been made accordingly. Referring to the specific points: 

Q1: The division into various categories presented in the tables seems rather arbitrary. It would profit from a more ordered approach. It is often not followed by text description. Many citations in the text are not found in the tables. Also I do not see the difference between scar segmentation and delineation (as presented separately in Table 2 and 3). But it refers to other tables too. Please reconsider the construction of the tables and their connection to text.
A1: We agree with the reviewer and we revised the papers listed in the tables, in agreement with the text.

Q2: Line 236 - for both LV and RV - usually it is far easier to apply AI to LV segmentation than RV analysis? Please discuss this problem separately. 

A2: We agree with the reviewer, briefly discussed separately and included appropriate references.

Q3: Line 157 - aortic stenosis is not a cardiomyopathy - please correct. Line 300 - normal contractile pump function refers to systolic heart failure and not all heart failure category (not diastolic) - please adjust accordingly.

A3: We have modified in agreement.

Q4: Please add a paragraph on current limitations of AI use in CMR and on the applications which are currently commercially available.

A4: We agree with the reviewer and have added a paragraph which briefly discusses current limitations. As regard the current commercially available applications, despite the appropriateness of the reviewer’s suggestion, we prefer not to discuss this point since it is beyond the scope of our paper and represents for as a conflict of interest element.

Q5: Line 141 - please remove F - there is no such panel on Figure 1.

Figure 6 label - assessment of...

Figure 7 - there are only panels A-E presented, but in the text panel F and G are also described - please correct

Line 284 - uin?

Line 294 - please remove capital letters from the disease names. Also please change ARVD to ARVC or AC throughout the text as it is the correct abbreviation, also in Line 296 there is only ARV

Line 308 - please use bold

A5: We have modified in agreement.

All the Best,

Andrea Igoren Guaricci, MD, PhD

Round 2

Reviewer 1 Report

The reviewer suggests accepting the manuscript in present form

Reviewer 2 Report

I have no further comments